# The Exosome-Mediated PI3K/Akt/mTOR Signaling Pathway in Neurological Diseases

**DOI:** 10.3390/pharmaceutics15031006

**Published:** 2023-03-21

**Authors:** Amin Iranpanah, Leila Kooshki, Seyed Zachariah Moradi, Luciano Saso, Sajad Fakhri, Haroon Khan

**Affiliations:** 1Pharmaceutical Sciences Research Center, Health Institute, Kermanshah University of Medical Sciences, Kermanshah 6734667149, Iran; 2USERN Office, Kermanshah University of Medical Sciences, Kermanshah 6715847141, Iran; 3Student Research Committee, Kermanshah University of Medical Sciences, Kermanshah 6714415153, Iran; 4Department of Physiology and Pharmacology “Vittorio Erspamer”, Sapienza University, P.le Aldo Moro 5, 00185 Rome, Italy; 5Department of Pharmacy, Abdul Wali Khan University, Mardan 23200, Pakistan

**Keywords:** exosome, neurological disease, neurodegenerative disease, targeted delivery, PI3K, Akt, mTOR

## Abstract

As major public health concerns associated with a rapidly growing aging population, neurodegenerative diseases (NDDs) and neurological diseases are important causes of disability and mortality. Neurological diseases affect millions of people worldwide. Recent studies have indicated that apoptosis, inflammation, and oxidative stress are the main players of NDDs and have critical roles in neurodegenerative processes. During the aforementioned inflammatory/apoptotic/oxidative stress procedures, the phosphoinositide 3-kinase (PI3K)/protein kinase B (Akt)/mammalian target of rapamycin (mTOR) pathway plays a crucial role. Considering the functional and structural aspects of the blood–brain barrier, drug delivery to the central nervous system is relatively challenging. Exosomes are nanoscale membrane-bound carriers that can be secreted by cells and carry several cargoes, including proteins, nucleic acids, lipids, and metabolites. Exosomes significantly take part in the intercellular communications due to their specific features including low immunogenicity, flexibility, and great tissue/cell penetration capabilities. Due to their ability to cross the blood–brain barrier, these nano-sized structures have been introduced as proper vehicles for central nervous system drug delivery by multiple studies. In the present systematic review, we highlight the potential therapeutic effects of exosomes in the context of NDDs and neurological diseases by targeting the PI3K/Akt/mTOR signaling pathway.

## 1. Introduction

Neurodegenerative diseases (NDDs) are progressive and chronic diseases characterized by imperceptible changes in neuronal structure [1,2]. NDDs are accompanied by a decrease in the population of specific neurons leading to disability, impaired normal functioning, dementia, and reduced life expectancy of patients. NDDs and neurological diseases are classified based on various parameters such as anatomical distribution, main clinical features, and important molecular abnormalities. Multiple sclerosis (MS), Parkinson’s disease (PD), Alzheimer’s disease (AD), amyotrophic lateral sclerosis (ALS), and Huntington’s disease (HD) are common NDDs/neurological diseases that result in the progressive deterioration of neurons [3,4]. The prevalence of neurodegenerative disorders is rapidly increasing alongside the increase in the aging global population. It is estimated that the number of dementia cases will increase from 13.5 million in 2000 to 36.7 million in 2050, which is alarming [5,6].

Although the pathogenesis of NDDs has not yet been precisely appreciated, numerous studies have underscored the undeniable contributions of inflammation, oxidative stress, apoptosis, and protein aggregation in some cases. The production of free radicals during both pathological and physiological processes plays a considerable role in several signaling pathways including phagocytosis, activation of enzymes, and regulation of the cell cycle. Excessive production of reactive oxygen species (ROS) causes various noxious effects such as protein and deoxyribonucleic acid (DNA) damage and lipid peroxidation [6,7,8,9]. As another contributor to the neurological disease, neuroinflammation brings about complex alterations in the brain’s immune system associated with multiple cellular and molecular aspects [6,7]. Such events lead to the alteration of glial cells, as well as augmentation of the concentration, activity, and levels of several inflammatory mediators including cytokines (e.g., interleukin-1β (IL-1β), IL-6, tumor necrosis factor-alpha (TNF-α)), chemokines (e.g., CCL2, CCL5, CXCL1), in addition to the generation of reactive nitrogen species (RNS) and ROS [10,11]. Increased permeability/breakdown of the blood–brain barrier (BBB), infiltration of peripheral immune cells, and edema are some of the other harmful processes that come to pass during neuroinflammation. Neuroinflammation contributes significantly to the development and progression of NDDs and neurological diseases. Therefore, suppression of inflammation could result in the prevention and amelioration of neurological disorders [6,12]. 

The phosphoinositide 3-kinase (PI3K)/protein kinase B (Akt)/mammalian target of rapamycin (mTOR) pathway is one of the key signaling pathways that plays pivotal roles in several pathological/physiological processes such as cellular migration, proliferation, apoptosis, and angiogenesis. In addition, according to numerous studies dysregulation of the PI3K/Akt/mTOR signaling pathway is associated with pathological effects in several disorders such as cardiovascular diseases [13,14], Crohn’s disease [15], cancer [16], and especially NDDs [9,12,17]. Following the activation of PI3K and phosphorylation of phosphatidylinositol 4,5- bisphosphate (PIP2), Akt is recruited to the cell membrane. As another key mediator, mTOR is associated with physiological neuroregeneration [18]. Moreover, the mTOR pathway is engaged in neuronal response-related signals in the gastrointestinal tract [19]. Additionally, mTOR signaling could be involved in autophagy where damaged mitochondria observed in the oxidative stress process could suppress PI3K signaling downstream through regulatory pathways such as the phosphatase and tensin homologue (PTEN) pathways [20]. Notably, PI3K/Akt pathway can regulate a wide range of upstream molecules such as growth factor receptors, G protein-coupled receptors (GPCRs), receptor tyrosine kinases (RTKs), extracellular signal-regulated kinase (ERK), and cytokines, which are involved in the attenuation of Janus kinase (JAK)/signal transducer and activator of transcription (STAT) activation [9,18,21,22,23,24]. Multiple studies have pointed to the significant effects of PI3K/Akt/mTOR signaling on several interconnected molecules implicated in oxidative stress (e.g., superoxide dismutase (SOD), heme oxygenase-1 (HO-1), ROS, catalase (CAT), nuclear factor erythroid 2–related factor 2 (Nrf2)), apoptosis (e.g., Bax/Bcl-2, and caspases), and inflammation (e.g., nuclear factor kappa B (NF-κB), ILs, matrix metalloproteinases (MMPs), chemokines, cytokines, and cyclooxygenase (COX)) [8,9,25,26,27]. 

Although a definitive treatment has still not been offered for NDDs, rapid fundamental advances in several new fields of science including nanotechnology, artificial intelligence, proteomics, stem cell therapy, genomics, gene therapy, exosome, and extracellular vesicles (EVs) technology combined with multidisciplinary approaches have opened new horizons for the treatment of NDDs and neurological diseases [28,29,30,31,32,33]. EVs are classified into exosomes (30–150 nm), microvesicles (50–1000 nm), or apoptotic bodies (800–5000 nm) based on their size and origin [34,35]. Exosomes originate from multi-vesicular bodies, and the budding of the plasma membrane is the main source of microvesicles and apoptotic bodies, which contain ribosomal ribonucleic acid (RNA), histones, and DNA produced from cells that undergo programmed cell death [33,36,37,38,39]. It has been demonstrated that microvesicles and exosomes play pivotal roles in intercellular communication as they are able to mediate long-distance transmission of biological information via transferring microRNAs (miRNAs), lipids, membrane receptors, RNA, and proteins between cells [33,36,37,38,39]. The type of cell and the physiological condition affect exosome composition, which may be strongly correlated with the development and progression of pathological processes [40]. As shown by recent studies, exosomes produced from cancer tissues can lead to disease progression. In addition, exosomes have been demonstrated to exert detrimental effects on neuronal tissues in neurological diseases. However, healthy cell-derived exosomes may possess therapeutic benefits. As a result, therapeutic strategies aimed to inhibit the production, uptake, or release of disease-promoting exosomes are of great promise [33,36,37,38,39]. There is already no review on the effects of exosome-mediated PI3K/Akt/mTOR signaling pathway in NDDs. This is the first systematic review that critically highlights the modulatory effects of exosomes as effective/safe drug delivery vehicles in the context of neurological diseases through PI3K/Akt/mTOR pathway.

## 2. Study Design and Methods

The current systematic review was performed based on Preferred Reporting Items for Systematic Reviews and Meta-Analysis (PRISMA) criteria. The keywords (“brain” OR “neuron” OR “Alzheimer’s disease” OR “dementia” OR “Parkinson’s disease” OR “multiple sclerosis” OR “spinal cord injury” OR “stroke” OR “depression” OR “aging” OR “seizure” OR “autism” OR “Amyotrophic lateral sclerosis” OR “ALS” OR “Huntington’s disease” OR “epilepsy”) AND (“PI3K” OR “phosphatidylinositol-3-kinase” OR “PKB” OR “Akt” OR “protein kinase B” OR “mTOR” OR “m-TOR” OR “mammalian target of rapamycin”) AND (“exosome”) were searched in [title/abstract/keywords] of the electronic databases, including Scopus, PubMed, and Web of Science. All pathways/factors related to PI3K/Akt/mTOR and exosomes were considered in the whole text. Data were collected without time limitation until September 2022. Only English language studies were included. Two independent researchers (S.Z.M. and A.I.) performed the search screening. Of the 786 articles collected via systematic search in aforementioned electronic databases, 276 and 222 articles were excluded due to being reviews and duplication, respectively. Moreover, 171 articles were excluded according to their title/abstract, 63 articles were excluded according to their full-text information, and 8 articles were excluded because they were not in English. Ultimately, 46 papers were included in this systematic review (Figure 1). In completing the search strategy, manual search of citations and reference lists falling within the authors’ expertise were also employed in the PI3K/Akt/mTOR signaling pathway as a pivotal therapeutic target in neurological diseases.

## 3. Neurodegenerative Diseases, Neurological Disorders, and Exosomes: Focusing on Pivotal Functions of PI3k/Akt/mTOR Signaling Pathway

Over the last few years, several studies demonstrated a promising future for EVs, especially exosomes for targeting neurological diseases. Concentrating on the pivotal PI3K/Akt/mTOR signaling pathway and associated factors, exosomes could combat AD, stroke, SCI, traumatic brain injury (TBI), ALS, optic nerve crush (ONC) injury, and other central nervous systems (CNS) injuries.

### 3.1. Exosomes and Alzheimer’s Disease, Cognition, Learning and Memory

AD is characterized as a common neurodegenerative disorder with a close relationship with dementia development. Reportedly, the prevalence of AD is conspicuous in the aged population and affects quality of life and social activities. In addition, AD pathology is associated with amyloid beta (Aβ) accumulation, tau protein hyperphosphorylation, neuroinflammation, and oxidative stress [41,42,43]. Based on the previous evidence, PI3K/Akt/mTOR axis is considered to be one of the most significant signaling pathways in the pathogenesis of NDDs. Indeed, this regulatory pathway displays critical functions in biological processes such as metabolism, cell proliferation, apoptosis, and angiogenesis [23]. Several lines of evidence have proposed that there is a correlation between exosome therapy and PI3K/Akt/mTOR signaling with neuronal damage. Reportedly, adipose mesenchymal stem cell-derived exosomes (ADSC-Exo) were shown to be effective in improving PC12 cell migration/proliferation and could repress apoptosis through boosting PI3K/Akt signaling pathway. In this line, ADSC-Exo treatment led to the overexpression of CD29, CD44, CD73, and CD105 as mesenchymal stem cell surface markers while reducing the expression of CD45 and HLA-DR [44]. In another study, bone marrow mesenchymal stromal cells (BMSCs)-Exo containing growth differentiation factor-15 (GDF-15) could exert a protective effect on Aβ_42_-induced SH-SY5Y cell injury through amelioration of the Akt/glycogen synthase kinase-3 beta (GSK-3β)/β-catenin signaling pathway. Of note, though this method of therapy promoted cell viability, it attenuated TNF-α, IL-6, IL-1β, IL-8 (as inflammatory cytokines), and apoptosis in Aβ_42_-induced SH-SY5Y cell damage [45]. In addition, a recent study illustrated that exosomes carrying curcumin (Exo-cur) significantly improved BBB crossing and ameliorated learning and memory deficits in okadaic acid (OA)-induced AD under both in vitro and in vivo models. Exo-cur also attenuated neural death and OA-induced tau hyperphosphorylation by stimulating the Akt/GSK-3β signaling pathway. Thus, Exo-cur represents a promising treatment through deactivation of microglia and mitigation of the OA-induced apoptosis of neuron cells. In addition, it led to improved neuronal function and alleviated AD symptoms [46]. According to an in vitro study, MSC-derived exosomal miR-223 (when applying 2 μg exosome-based on exosomal protein content per 1 × 10^5^ recipient cells) could target PTEN and stimulated PI3K/Akt signaling pathway in an in vitro model of AD. Thus MSC-derived exosomes can increase cell migration and decrease neuronal apoptosis and inflammatory mediators including IL-6, IL-1β, and TNF-α [47]. Other studies suggested that neural stem cell (NSC)-derived exosomes (NSC-Exo) prevented high-fat diet (HFD)-dependent memory deficits in male C57BL/6 mice by restoring the cAMP response element-binding protein (CREB)/brain-derived neurotrophic factor (BDNF)/tropomyosin receptor kinase B (TrkB) signaling and the expression of synaptic plasticity-associated genes. Taken together, treatment with NSC-Exo (1.5 μg per nostril, three times per week) could upregulate CREB/BDNF/TrkB signaling in the hippocampus of HFD mice, underscoring the treatment as a potential therapy for metabolic disease-related cognitive impairment [48]. Furthermore, it was shown that MSCs-miR-132-3p-Exo improved cognitive decline and synaptic dysfunction as well as promoted dendritic spine density and neuron numbers by activating the Ras/Akt/GSK-3β pathway in vascular dementia under in vitro and in vivo models [49].

Overall, exosomes play critical roles in circumventing cognitive/memory dysfunction through affecting the PI3K/Akt/mTOR pathway and its associated markers (Table 1). In summary, although more in vitro and in vivo studies are necessary to accurately prove the role of exosomes in AD, according to the mentioned reports, exosomes could be regarded to be a promising candidate for the prevention or treatment of AD, cognition, learning, and memory deficit. Additionally, exosomes could be used in combination with other drugs, which requires comprehensive pre-clinical and clinical studies. Future clinical applications should also focus on the usefulness of exosomes in predicting the emerging symptoms of AD. Additionally, application of the microfluidic technique will show the road to diagnosis of the primary symptoms of AD before the late stages of the disease. High biocompatibility, high BBB penetration, long blood circulation, prevention of degradation, and tissue targeting are additional advantages of exosomes to be used in AD [50]. Due to the capacity of exosomes in RNA transport, stability, and their BBB-crossing capability, exosomes are appropriate carriers in combating AD [51]. Additionally, neuron-derived exosomes could make Aβ conformational modifications to non-toxic fibrils and cause increased microglia uptake [52]. Altogether, exosomes are promising drug/enzyme/miRNA delivery vehicles, and also play a critical role in scavenging waste neurotoxic agents. The neuroprotective potential of exosomes is also closely linked with their ability to block NF-κB.

### 3.2. Exosomes and Stroke

Stroke is considered the second leading cause of mortality and the third leading cause of disability worldwide with an increasing growth globally [68,69]. As a socioeconomic problem, stroke has a high morbidity and mortality rate [70]. Thus, it could decrease the life quality of patients and impose high economic costs on patients and healthcare systems [69]. Its occurrence is predicted to reach 23 million people in the world by 2030 [71]. There are three different types of stroke. Ischemic stroke and hemorrhagic stroke are the major types of strokes [69,70]. A transient ischemic attack (TIA) or mini-stroke is another type of stroke in which symptoms last less than 24 h and can be a warning sign for future strokes [68]. Ischemic stroke or brain ischemia is the most common type of stroke and occurs due to a blockage of blood flow to the brain. Obstruction is usually caused by blood clots and results in hypoxia, nutrient deprivation, and the induction of inflammation and oxidative stress [69,70,72]. Hemorrhagic stroke is caused by a blood vessel rupture, leading to intracerebral hemorrhage (ICH) or subarachnoid hemorrhage (SAH) [73]. 

The persistent bleeding from a hemorrhagic stroke causes oxidative stress, neuroinflammation, apoptosis, and BBB disruption [56,70]. In addition, the PI3K/Akt/mTOR pathway can modulate multiple cellular and molecular events including oxidative stress factors, inflammatory responses, programmed cell death, and cell survival, and it protects neurons and the brain from ischemic damages [9,74,75]. Hence, this pathway can be considered as one a significant signaling pathway and therapeutic target for neuroprotection against stroke.

Xin et al. illustrated that MSCs-miR-17-92^+^-Exo enhanced PI3K/Akt/mTOR activation by reducing PTEN expression, which modulated the stroke injury induced by middle cerebral artery occlusion (MCAO) in rats. In addition, they demonstrated that Exo-miR-17-92^+^ significantly elevated corticospinal tract (CST) axonal remodeling, myelination, neurological recovery, and reversed MCAO-induced behavioral dysfunctionality [53]. In another similar model, Exo-miR-17-92^+^ represented protective effects on brain injury and increased functional recovery, axonal density, neurogenesis, oligodendrogenesis, and spine and dendritic plasticity through inhibition of PTEN activity and increases in p-Akt, p-mTOR, and p-GSK-3β activity [54]. Healthy rat serum-derived exosomes have also shown neuroprotective effects in combating stroke injury in in vitro (50 μg/mL) and in vivo (800 μg/kg, i.v.) models through elevation of the p-Akt/Akt ratio, claudin-5, zonula occludens (ZO)-1, the Bcl-2/Bax ratio, and sequestosome 1 (SQSTM1)/p62 expression while reducing BBB leakage, cell apoptosis, MMP-9, cleaved caspase-3, and LC3B-II/LC3B-I ratio. Healthy rat serum-derived exosomes also improved the results of behavioral tests [55]. MSCs-miR-132-3p-Exo showed a notable amelioration of infarct volume, BBB dysfunction, neurological deficit scores (NDS), brain edema, and injury in focal ischemic stroke induced by transient MCAO in C57BL/6 mice through suppressing the expression of RASA1, reducing apoptosis and the ROS levels, and upregulating Ras, ZO-1, claudin-5, and the Ras/PI3K/Akt/endothelial nitric oxide synthesis (eNOS) pathway [56].

In a rat model of ICH, Duan and colleagues showed that BMSC-miR-146a-5p-Exos ameliorated neurological function through decreasing oxidative stress and inflammatory mediators interconnected to the PI3K/Akt/mTOR pathway including COX-2, malondialdehyde (MDA), inducible nitric oxide synthase (iNOS), TNF-α, myeloperoxidase (MPO)-positive cells, monocyte chemoattractant protein-1 (MCP-1), IL-1β, and IL-6 and suppression of microglial M1 polarization. These effects were associated with the downregulation of nuclear factor of activated T cells 5 (NFAT5) and IL-1 receptor-associated kinase 1 (IRAK1) expression [57]. In another rat model of ICH, MSC exosome-transferred miR-133b (100 μg via the tail vein, i.v., 72 h after ICH) reduced neuronal apoptosis and neurodegeneration by RhoA downregulation and ERK1/2/CREB pathway activation [58]. A recent report by Wang and colleagues showed that ADSC-Exos could decline neurological deficits and promote cell viability through suppression of insulin-like growth factor-binding protein 5 (IGFBP5) and increase in the expression of PI3K/Akt signaling pathway components in in vitro and in vivo models of SAH [59]. Recent studies have shown that BDNF could confer protection against apoptosis and neuronal injury by activating the ERK and/or PI3K/Akt pathway to stimulate the phosphorylation of CREB [76,77]. In another study, miR-206-knockdown exosomes from human umbilical cord-derived mesenchymal stem cells (hucMSCs) demonstrated significant in vivo therapeutic effects in SAH-induced early brain injury (EBI) through the BDNF/TrkB/CREB signaling pathway. BDNF/TrkB/CREB signaling pathway activation inhibited neuronal death and minimized brain edema and neurological dysfunction. In addition, hucMSCs-derived miR-206-knockdown exosomes ameliorated Bcl-2/Bax ratio and reduced cleaved caspase-8 induced by brain injury [60].

Stem cell-derived exosomes (SC-Exos) attenuated neuronal apoptosis, augmented interferon-gamma (IFN-γ) and Bcl-2, and decreased IL-1α, IL-2, TNF-α, Bax, cytochrome C (CytC), and cleaved caspase-3 and caspase-9 production in rats with cerebral ischemia/reperfusion (I/R) injury. These mechanisms could be linked to PI3K/Akt pathway-mediated mitochondrial apoptosis [61]. Wu et al. elucidated the in vitro and in vivo neuroprotective effects of astrocyte-derived exosome (ATC-Exos)-contained miR-34c in terms of the protection of Neuro 2A (N2a) mouse neuroblastoma cells by increasing toll-like receptor 7 (TLR7) expression and downregulating the NF-κB/mitogen-activated protein kinase (MAPK) pathways. It has been shown that treatment with ATC-Exos significantly reduces infarction volume; brain edema; inflammatory mediators including IL-6, IL-8 and TNF-α; apoptotic factors such as Bax, cleaved caspase-3, and cleaved PARP; and ameliorates neurological deficits caused by cerebral I/R injury [62]. Furthermore, 10 μg/mL BMSC-Exos conferred protective effects on oxygen–glucose deprivation/reperfusion (OGD/R)-induced injury in PC12 cells and promoted cell viability by targeting the AMP-activated protein kinase (AMPK)/mTOR pathway [63]. In another study, Bu and colleagues showed the neuroprotective advantages of ATC-Exos contained miR-361 under in vitro (30 μg/mL) and 2 mL exosomes (30 μg/mL) via the caudal vein, twice a week for 2 weeks) models of cerebral I/R injury through attenuation of the AMPK/mTOR signaling pathway via targeting cathepsin B (CTSB). In addition, infarct volume, cerebral edema, cleaved poly adenosine diphosphate-ribose polymerase (PARP), cleaved caspase-3, and Bax exhibited a significant decrease in the ATC-Exo-treated groups, while neuronal viability increased [64].

In addition, BMSC-derived exosomal miR-29b-3p [65], exosomes from different H9 human embryonic stem cell (hES)-derived cells [66], and miR-126 enriched endothelial progenitor cells (EPCs)-released exosomes [67] are among other exosomes with promising protective activities in combating brain injury through modulation of the PI3K/Akt/mTOR signaling pathway and the related mediators.

Altogether, the above-mentioned studies highlight the promising protective effects of exosomes against different types of stroke by employing different mechanisms such as the modulation of inflammatory cytokines, autophagic molecules, and oxidative and apoptotic factors. These functions of exosomes typically pass through the PI3K/Akt/mTOR signaling pathway (Table 1). In addition, these nano-sized structures display different advantages such as low immunogenicity and high BBB penetration capacity over other therapeutic modalities. However, there exist several challenges against exploiting EVs for therapeutic applications including drug interactions with EV components, a lack of controlled drug release mechanisms, and a lack of specific biomarkers. Exosome therapy still has various limitations and more studies are needed to find ways to elevate their circulation half-life, increase the quantity of bioactive molecules loaded in exosomes, enhance their stay at the disease site, and use them for targeted delivery to highlight their eligibility in clinical trials to combat stroke. Thus, further studies including extensive in vitro and in vivo experimentations as well as comprehensive pre-clinical and clinical trials on exosomes are necessary to introduce exosomes as potential agents for modulating stroke. Multiple in vitro and in vivo reports have proven that exosomes could increase functional recovery, angiogenesis, neurovascular remodeling, and synaptic plasticity and could be neurorestorative after stroke through transfer of different types of cargoes such as miRNAs, proteins, lipids, and phytochemicals [78,79]. 

### 3.3. Exosomes and Spinal Cord Injury

SCI is a critical insult to the spinal cord that causes temporary or permanent motor and sensory impairment and disability [80,81]. SCI affects most of the body’s functions and can diminish patients’ quality of life [80]. SCI is classified as non-traumatic or traumatic SCI, according to its etiology [82]. Recent studies have shown that the PI3K/Akt/mTOR pathway plays critical roles in the recovery of the spinal cord after injury via regulation of the release of proinflammatory cytokines, oxidative stress, cell death, neuron growth, differentiation, and formation of glial scar [75,83]. Thus, it is critical to consider the roles of exosomes in the context of SCI via modulation of the PI3K/Akt/mTOR signaling pathway.

In 2021, Chen et al. investigated the advantages of BMSC-miR-26a-Exos in SCI in in vitro and in vivo models. The results demonstrated that interference of the PTEN/Akt/mTOR pathway is the major neuroprotective mechanism governing BMSC-miR-26a-Exo-mediated functional recovery, neurogenesis, axonal regeneration, and attenuation of astrocyte inflammation, autophagy, and glial scarring [84]. In another report, miR-338-5p overexpressing BMSC-derived exosomes showed neuroprotective activities by elevating neuronal survival, modulating oxidative stress factors, and suppressing SCI-induced cell death in both in vitro and in vivo experiments. These effects were attributed to the PI3K/Akt pathway via downregulation of cannabinoid receptor 1 (Cnr1) and cAMP-mediated Rap1 activation [85]. Furthermore, exosome-shuttled miR-216a-5p from hypoxic BMSCs (200 μg/mL) was shown to deviate microglia/macrophage polarization from M1 pro-inflammatory phenotype to M2 anti-inflammatory phenotype, increase IL-4, and IL-10, and decrease iNOS, TNF-α, IL-1β, and IL-6 through activation of the PI3K/Akt pathway and TLR4/NF-κB pathway suppression. These pathophysiological signaling pathways led to improved functional, gait, and motor recovery in a C57BL/6 mice model of SCI [86]. Luo and colleagues showed that exosomes from G protein-coupled receptor kinase 2 interacting protein 1 (GIT1)-overexpressing BMSCs promoted neural regeneration, functional behavioral recovery, and antiapoptotic factors (e.g., Bcl-2) while reducing glial scar formation, inflammatory mediators (e.g., IL-1β, IL-6, and TNF-α), and proapoptotic factors (e.g., Bax and cleaved caspase-3 and -9) which led to the alleviation of apoptosis and neuroinflammation as evaluated by in vitro and in vivo experiments. Upregulation of the PI3K/Akt signaling pathway could be presumed to be one of the major protective mechanisms adopted by these exosomes in combating traumatic SCI [87]. 

Wang et al. showed that MSCs-Exo conferred neuroprotection by anti-inflammatory activities through downregulation of the nuclear translocation of NF-κB p65, TNF-α, IL-1α, IL-1β, and p-IKBα as demonstrated by in vitro and in vivo experiments [88]. In another study, MSCs-miR-126-Exo enhanced neurogenesis, angiogenesis, functional recovery, connectivity value, and blood vessel numbers and diminished apoptosis and lesion volume after SCI. Such effects were mediated through inhibition of sprouty-related EVH1 domain-containing protein 1 (SPRED1) and PI3K regulatory subunit 2 (PIK3R2) [89]. Of other reports on the exosome-mediated regulation of PI3K/Akt/mTOR, neuron-derived exosomes transmitting miR-124-3p could remarkably attenuate axonal damage, lesion volume, M1 microglia and A1 astrocytes activation. In addition, it minimized pro-inflammatory cytokines (TNF-α, IL-1α, IL-6, and IL-1β), iNOS, and improved functional/gait recovery through the regulation of PI3K/Akt/NF-κB signaling pathways as assessed by in vitro and in vivo experiments [90]. Chen et al. investigated the therapeutic effect of FTY720-loaded exosomes derived from nerve stem cells (NSCs) (FTY720-NSCs-Exos) in SCI using in vitro and in vivo models. In this line, FTY720-NSC-Exos increased p-Akt, Bcl-2, claudin-5, ZO-1, and locomotor function while reducing PTEN, SCI lesion, edema formation, inflammatory cell infiltration, and apoptosis of neuronal cells via regulation of the PTEN/Akt pathway which led to neuroprotective effects [91]. Pan and colleagues demonstrated that primary Schwann cell-derived exosomes (SCDEs) could promote recovery after SCI through modulation of NF-κB/PI3K [92] and epidermal growth factor receptor (EGFR)/Akt/mTOR signaling pathways as elucidated by in vitro and in vivo assays [93]. Human urine stem cell (HUSC)-ANGPTL3-Exo exhibited a neuroprotective activity against SCI and increased angiogenesis, spinal cord regeneration, sensory improvement, vessel volume fraction, and recovery of neurological functional by interference with the PI3K/Akt signaling pathway as confirmed by in vitro and in vivo experiments [94]. According to in vitro and in vivo evidence, hucMSCs-miR-199a-3p/145-5p-Exos diminished neurological symptoms via decreasing the inflammation and apoptotic cells and increasing TrkA, p-Akt, and p-Erk, and interconnected to nerve growth factor (NGF)/TrkA pathway [95]. Additional studies discovered that peripheral macrophage (PM)-Exos [96], resveratrol-primed exosomes secreted by primary microglia [97], and pericyte exosomes [98] possessed neuroprotective properties in combating SCI through PI3K/Akt/mTOR signaling pathway and related factors.

Therefore, exosomes could be contemplated as possible therapeutic tools for SCI by modulating neuroapoptosis, neuronal oxidative stress, and neuroinflammation through the PI3K/Akt/mTOR signaling pathway. Considering the challenges mentioned for exosomes, it seems that more in vivo studies are needed to conduct clinical trials to determine the role of exosomes in neurological diseases such as SCI. Taken together, these reports suggest exosomes as new tools for modulating SCI. As natural carriers of biologically active cargoes, exosomes could not only remarkably ameliorate functional recovery, neural regeneration, and angiogenesis of animals with SCI, but also notably elevate the expression of antioxidant factors, anti-apoptotic protein Bcl-2, and anti-inflammatory mediators including IL-4, IL-10, and IL-13. Exosomes markedly reduced pro-inflammatory factors such as IL-1β, IL-6, and TNF-α and the expression of the apoptotic protein Bax. These effects are in a near linkage with their ability to regulate the PI3K/Akt/mTOR signaling pathway.

Overall, because exosomes can effectively cross the BBB, they could be used for the treatment and diagnosis of several neurological disorders, such as SCI. In addition, more studies are required to clarify the specific role of exosomes in SCI and bring hope for clinical treatment of SCI.

### 3.4. Exosomes and Traumatic Brain Injury

TBI is defined as a mechanical injury to the parenchymal tissues and meninges of the brain associated with inflammatory and oxidative responses [99,100]. Based on in vitro and in vivo studies, endothelial colony-forming cells (ECFCs)-derived exosomes rescued the expression of tight junction (TJ) proteins by targeting the PTEN/Akt pathway. Indeed, theses exosomes decreased PTEN expression and activated Akt phosphorylation. On the other hand, pretreatment with exosomes showed beneficial effects in declining MMP-9 expression, Evans blue dye extravasation, and TJ protein degradation in mice with TBI [101]. To suppress TBI-related neuroinflammation, microglial exosomes-contained miR-124-3p could be considered as a potential therapeutic molecule in preventing neuronal inflammation following TBI through targeting phosphodiesterase 4B (PDE4B) and ultimately repressing the mTOR signaling pathway. Furthermore, elevated miR-124-3p in microglial exosome-mediated inhibition of neuronal inflammation was associated with improved anti-inflammatory M2 polarization of microglial cells [102]. Growing evidence has indicated that hADSC-Exo shows neuroprotective effects against in vivo TBI models via suppression of the classical NF-κB and MAPK signaling pathways to restrain microglia/macrophage activation. Overall, hADSC-Exo administration caused sensorimotor functional recovery in TBI rats, improved hippocampal neurogenesis, suppressed neuroinflammation, and mitigated neuronal apoptosis. Notably, in vitro application of hADSC-Exo markedly inhibited M1 microglial polarization and increased M2 microglial polarization. Hence, intracerebroventricular hADSC-Exo administration could serve as a precious therapeutic modality against CNS diseases [103]. Further in-depth studies are needed to confirm the effectiveness of exosomes in TBI. Collectively, exosomes have exhibited a promising future in combating TBI through modulating the PI3K/Akt/mTOR pathway (Table 2).

Exosomes could be considered powerful biomarkers for the diagnosis of TBI. Exosome therapies are effective approaches for improving neurological and functional recovery via increasing neurite growth and neurogenesis by delivery of gene or pharmacological agents after TBI [104,105]. Moreover, there are still some challenges with the design, separation, and purification procedure of exosomes [106]. Therefore, future research should explore a cheap, quick, and simple standardized method for the generation of exosomes and establish the best strategies for exosome modulation of TBI toward promoting the translation of preclinical reports outcomes to clinical studies. In conclusion, the evidence shows that exosomes might have great potential in neurorestorative therapy for TBI.

### 3.5. Exosomes and Other Neurological Disorders

It is worth mentioning that various types of exosomes could ameliorate several other neurological diseases via the PI3K/Akt/mTOR pathway and other interconnected signaling pathways [107]. Emerging reports suggest that the PI3K/Akt signaling pathway and the associated proteins can be targeted by ADSC-Exo administration in ALS. Notably, it has been reported that treatment with adipose stem cell (ACS) exosomes reduces pro-apoptotic proteins such as cleaved caspase-3 and Bax while increasing the anti-apoptotic protein Bcl-2 in an in vitro model of ALS. In addition, Western blot analysis revealed increased p-Akt and SOD1 expression in ASC exosome-treated cells [108]. Moreover, another study provided evidence on the effects of fibroblast-derived exosomes (FD-Exo) (50 ng/mL) on the promotion of axonal regeneration in the injured CNS by recruiting Wnt10b toward lipid rafts and subsequently activating mTOR signaling through GSK-3β and tuberous sclerosis complex 2 (TSC2) [109]. In an experimental study, researchers found that FD-miR-673-5p-Exo is associated with peripheral neuron myelination of Schwann cells via stimulation of the TSC2/mTOR complex 1 (mTORC1)/sterol-regulatory element binding protein 2 (SREBP2) axis. Indeed, FD-miR-673-5p-Exo can enhance peripheral neuron myelination in newborn rats and myelin gene expression in Schwann cells [110]. In this context, MSC-exosomes caused improved axonal growth of cortical recipient neurons and increased the length of distal axons through augmenting p-mTOR and p-GSK-3β and reducing PTEN; it was also found that elevation of miR-17-92 cluster in the exosomes (miR-17-92 exosomes) further promoted axonal growth [111]. According to the reported studies, exosomes derived from ADSCs (50 ng/mL) show inhibitory effects on lipopolysaccharide (LPS)-induced injury in SH-SY5Y and BV-2 cells by reducing TNF-α, IL-1β, IL-6, COX-2, iNOS, p-P38, p-P65, p-ERK, and p-JNK. In addition, ADSC-exosomes mitigated LPS cytototoxic effects and suppressed neuroinflammation by repressing the NF-κB and MAPK signaling pathways. In conclusion, ADSC-exosomes reveal therapeutic effects on neural injury induced by microglia activation [112]. Evidence indicates that exosomes released by human ADSC (hAMSCs) seem to be a promising therapeutic approach against neural injury induced by glutamate in PC12 cells by upregulating PI3K/Akt signaling pathway [113]. Based on the research, MSC-Exo in a rat model of ONC injury decreased the expression of pro-inflammatory cytokines such as IL-1β, IL-6, IL-8, TNF-α, and MCP-1, whereas it increased the anti-inflammatory mediator IL-10. In addition, MSC-Exo decreased the ONC-induced apoptosis of the retinal ganglion cells (RGCs) via promotion of the Bcl-2/Bax ratio and reduction in caspase-3 activity, and Akt phosphorylation/activation was observed following intravitreal MSC-Exo administration [114]. Altogether, the authors demonstrated that exosomes could serve as potential therapies in the management/treatment of other neurological diseases via regulating PI3K/Akt/mTOR and associated signaling pathways (Table 3).

## 4. Conclusions and Future Perspective

A growing number of studies have highlighted the pivotal role of the PI3K/Akt/mTOR signaling pathway in the CNS and associated dysregulations. In addition, recent reports are demonstrating the modulatory roles of PI3K/Akt/mTOR pathway and the related inflammatory mediators (e.g., NF-κB, TNF-α, ILs, CRP, COX-2, and MMP-9); oxidative/antioxidative factors (e.g., GSH, SOD, CAT, Nrf2/HO-1, and iNOS); apoptotic factors (e.g., Bax, Bcl-2, caspase-3, and caspase-9); and the interlinked pathways (e.g., MAPK, CREB/BDNF, GSK-3β, ERK1/2, and JAK/STAT) in the progression/treatment of NDDs and other neurological diseases (Figure 2). Considering the side effects and resistance mechanisms to conventional neuroprotective drugs, it is of great importance to provide alternative therapies in combating neurological diseases. Exosomes can be considered to be promising therapies owing to their specific properties including low immunogenicity, flexibility, high BBB penetration capacity, and the capability of being used as drug delivery vehicles that could provide a long-lasting concentration of medication in the CNS while having few side effects and especially regulating the aforementioned inflammatory, oxidative, and apoptotic pathways/factors. To combat such pathophysiological mechanisms, it is necessary to consider novel approaches to drug delivery into the CNS using natural entities with fewer side effects. Exosomes strategically carry drugs and possess suitable stability and half-life. Understanding the roles that exosomes play in communication between various cell types at diverse sites provides a critical step forward in revealing the details of cell communication [115]. However, exosome-based therapeutic strategies and research progress in the field of exosomes face some challenges including difficulties in characterization, lack of controlled drug release mechanisms, inefficient isolation methods, drug interaction with exosome/EV components, and lack of specific biomarkers. Exosomes are suitable carriers for various cargoes including proteins, drugs, and natural products. There are now limitations in the manipulation of exosomes to be used in diseases. Separation, identification, and diagnosis of exosomes are critical steps requiring future research [50]. The optimization of operational procedures is necessary, and the characterization of exosome cargoes’ mediating therapeutic effects is warranted. New low-cost techniques to obtain a large amount of high-purity exosomes need to be provided. Furthermore, increasing the half-life of exosomes and targeting the ability of exosomes could provide clinical-grade exosomes as promising therapeutic approaches for future studies [116].

Accordingly, considering the critical role of PI3K/Akt/mTOR in NDDs and the multiple advantages of EVs in the context of effective and safe drug delivery, exosomes could confer a significant neuroprotective role in combating AD, PD, ALS, stroke, TBI, and other neuronal disorders [117].

This is the first systematic review with a focus on the pivotal role of PI3K/Akt/mTOR pathway targeted by exosomes in NDDs. We critically highlighted the modulatory functions of exosomes in NDDs through PI3K/Akt/mTOR pathway. Future investigations should incorporate extensive in vitro and in vivo experimentations and well-designed randomized clinical trials on exosomes to clarify in more detail the crucial roles of the PI3K/Akt/mTOR pathway and the modulatory effects of exosomes in the context of NDDs.

## Figures and Tables

**Figure 1 pharmaceutics-15-01006-f001:**
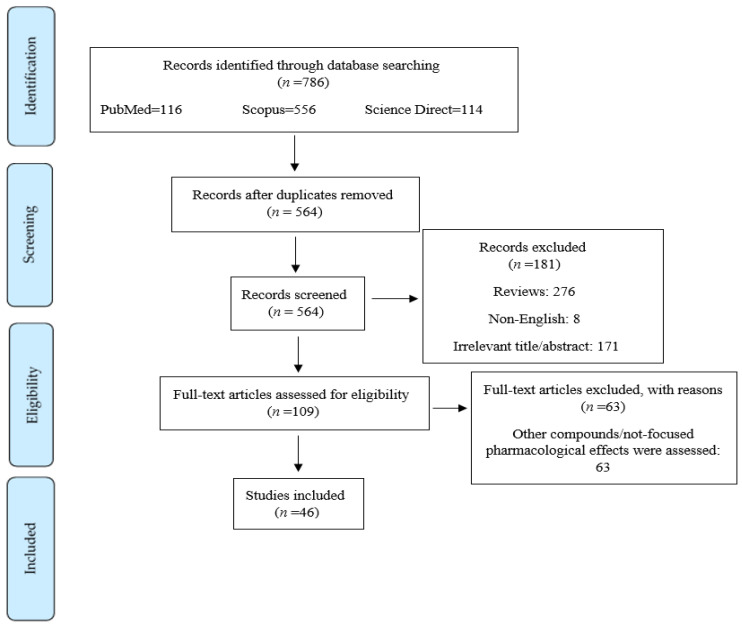
Flowchart of the process of literature search and selection of relevant reports.

**Figure 2 pharmaceutics-15-01006-f002:**
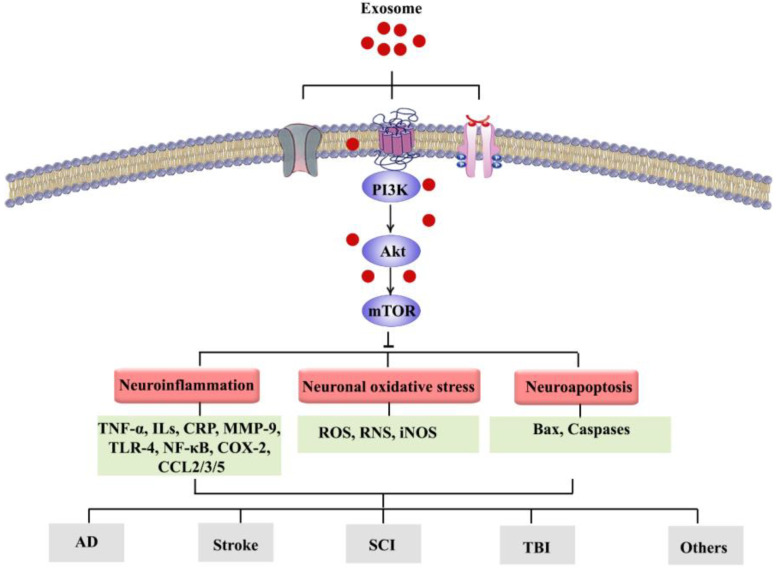
Exosomes can target the PI3K/Akt/mTOR signaling pathway, thereby modulating various interconnected events involved in NDDs towards therapeutic application.

**Table 1 pharmaceutics-15-01006-t001:** Exosomes circumvent AD and stroke via PI3K/Akt/mTOR and associated pathways.

Source of Exosomes	Cargo or Intermediate Molecule	Disease	Method (In Vitro/In Vivo)	Dose/Route of Administration	Mechanism of Actions and Outcomes	References
ADSC	_____	AD	in vitro: PC12 cells	10, 50 and 100 μg/mL	↑p-PI3K, p-Akt, CD29, CD44, CD73, CD105, cell proliferation, and migration; ↓CD45, HLA-DR, and cell apoptosis	[44]
BMSCs	GDF-15	AD	in vitro: SH-SY5Y cells	_____	↑Cell viability, Akt/GSK-3β/β-catenin pathway; ↓Apoptosis, TNF-α, IL-6, IL-1β, and IL-8	[45]
Macrophage cell line (RAW 264.7)	Curcumin	AD	in vivo: Okadaic acid-induced AD in C57BL/6 mice; in vitro: hCMEC/D3 cells	0.4 mg/kg, single i.v. dose, and 100 μg/mL, i.p. for 7 days	↑Curcumin solubility, stability, bioavailability, cellular uptake and BBB-crossing, p-Akt, p-Ser9 GSK-3β; ↓Learning deficiencies, cognitive decline, escape latency, cell apoptosis, tau hyperphosphorylation, p-Ser396 tau, neuronal injury, Bax and c-caspase 3;	[46]
HucMSCs	miR-223	AD	in vitro: Aβ_1–40_-induced injury in SH-SY5Y cells under hypoxic conditions	2 μg per 1 × 10^5^ recipient cells	↑p-Akt; ↓Apoptosis, scratch area, IL-6, IL-1β, TNF-α, CRP, and PTEN	[47]
NSC	_____	HFD-related cognitive decline	in vivo: male C57BL/6 mice	1.5 μg per nostril, 3 times per week	↑BDNF, nNOS, Sirt1, Egr3, RelA, and pTrkB; ↓Memory impairment	[48]
MSCs	miR-132-3p	VD	in vivo: VD-induced in male C57BL/6 mice; in vitro: OGD-injured neurons	1 × 10^10^ particles/100 μL, via the tail vein once every 7 days for 21 days	↑Cognitive function, neuron number, dendritic spine density, synaptic plasticity, Ras, p-Akt, p-GSK-3β, and neurite elongation; ↓Aβ, p-tau, RASA1, and apoptosis	[49]
MSCs	miR-17-92	Stroke	in vivo: 2 h intraluminal filament-induced MCAO in Wistar rats; Ex-vivo rat organotypic brain slice culture model	3 × 10^11^ particles/rat, i.v.	↑GAP-43 immunoreactivity, cortical and intracortical axonal density, myelin density, neuronal plasticity, contralesional axon number and total length, CST axonal remodeling, and functional recovery ↓Time to remove the adhesive tabs, mNSS score, and lowest threshold value of ICMS; ↑MBP^+^ myelin and NFH^+^, PTEN/PI3K/Akt/mTOR pathway	[53]
MSCs	miR-17–92	Stroke	Wistar rats were subjected to 2 h intraluminal filament-induced MCAO	100 μg total exosome protein or 3 × 10^11^ particles per rat, i.v.	↑Functional recovery, axonal density, p-NF-H immunoreactive area, synaptophysin immunoreactivity, primary and secondary neurite branching, spine density, dendritic plasticity, neurogenesis, oligodendrogenesis, p-Akt, p-mTOR, and p-GSK-3β; ↓mNSS score, number of Foot-fault and PTEN level	[54]
Healthy rat serum-derived exosomes	_____	Stroke	in vivo: Focal cerebral ischemia induced by the intraluminal suture MCAO method in SD rats in vitro: OGD/R injury model in bEnd.3 immortalized mouse brain endothelial cells	800 μg/kg, i.v. 50 μg/mL	↑Neurobehavioral scores, total moving distance, neuronal spine density, claudin-5, ZO-1, Bcl-2/Bax ratio, p-Akt/Akt and SQSTM1/p62 expression, ↓Infarct volumes, %distance and %time in center, BBB leakage, Evans blue dye extravasation, MMP-9, cleaved caspase-3 and LC3B-II/LC3B-I ratio; ↑SQSTM1/p62 expression, ↓Apoptotic cells, TUNEL^+^/CD31^+^ cells, cleaved caspase-3 and LC3B-II/LC3B-I ratio	[55]
MSCs	miR-132-3p	Ischemic stroke	in vivo: Focal ischemic stroke induced by transient MCAO in C57BL/6 mice in vitro: H/R injury model in mouse brain microvascular endothelial cells	1 × 10^10^ particles/100 μL in PBS via the tail vein, i.v. 50 μg/mL	↑cMVD and CBF;↓ROS, apoptosis, Evans blue dye extravasation, brain water content, infarct volume, NDS, and BBB disruption; ↑Ras, p-PI3K/PI3K, p-Akt/Akt and p-eNOS/eNOS, ZO-1, and Claudin-5; ↓RASA1, ROS, paracelluar permeability and apoptosis	[56]
BMSCs	miR-146a-5p	ICH	in vivo: Collagenase type IV-induced ICH in male SD rats, by an intrastriatal injection	100 μg/mL, 100 μg via the tail vein, i.v.	↑SOD and neurological function; ↓Microglia M1 polarization, iNOS, COX-2, MCP-1, IRAK1, NFAT5, TNF-α, IL-1β, IL-6, MPO-positive cells, MDA, OX42-positive cells, Iba-1^+^/MHC-II^+^, apoptotic, and degenerative neurons	[57]
MSCs	miR-133b	ICH	in vivo: An autologous arterial blood ICH model in adult male SD rats	100 μg via the tail vein, i.v., 72 h after ICH	↑p-ERK1/2/ERK1/2 and p-CREB/CREB; ↓RhoA expression, neuronal apoptosis and neurodegenerative neurons	[58]
ADSCs	miR-140-5p	SAH	in vivo: SAH-induced neurological dysfunction in rat in vitro: TDP-43-induced neuronal injury	_____	↑Cell viability and PI3K/Akt activation; ↓IGFBP5 expression and apoptosis	[59]
HucMSCs	miR-206-knockdown	SAH	in vivo: SAH-induced EBI in double blood injection model in SD rats	200 μL PBS containing 400 μg exosomes, i.v. injected into the femoral vein 1 h after SAH	↑Bcl-2, BDNF, TrkB, and p-CREB; ↓Bax, caspase-8, neurological deficit, brain edema, and neuronal apoptosis	[60]
SC	_____	Cerebral I/R injury	in vivo: Focal cerebral I/R induced by the improved Longa method in rats	_____	↑Latency for the novel arm, IFN-γ, and Bcl-2; ↓Novel entries, IL-1α, IL-2, TNF-α, Bax, cleaved caspase-3, cleaved caspase-9, CytC, PI3K, Akt, and neural cell apoptosis	[61]
ATCs	miR-34c	Cerebral I/R injury	in vivo: Wistar rats MCAO model in vitro: OGD/R model in N2a mouse neuroblastoma cells	20 and 30 μg/mL via the tail vein after ischemia, 20 μM	↑Nissl bodies and c-fos positive cell numbers; ↓Neuronal injury, NDS, infarct volume, brain water content, IL-6, IL-8, and TNF-α; ↑Cell proliferation, EdU positive cell index, and TLR7; ↓Bax, cleaved caspase-3, cleaved PARP, apoptosis, and NF-κB/MAPK axis	[62]
BMSCs	_____	Cerebral I/R injury	in vitro: OGD/R model in PC12 cells	10 μg/mL	↑Cell viability, autophagic flux, p-AMPK/AMPK; ↓LDH, morphological changes, pyroptosis, NLRP3, ROS, cleaved caspase-1, IL-1β, GSDMD-N, p-mTOR/mTOR, and P62	[63]
ATCs	miR-361	Cerebral I/R injury	in vivo: Wistar rats reversible MCAO model in vitro: OGD/R model in PC12 mouse neuroblastoma cells	2 mL exosomes (30 μg/mL) via the caudal vein, twice a week for 2 weeks 30 μg/mL	↑Nissl bodies, C-fos and neuronal viability; ↓Nerve damage, NDS, brain water content, infarct volume, cerebral edema, apoptosis, AMPK and mTOR mRNA and protein levels; ↑Cell activity, cell proliferation, and EdU positive cell index; ↓Apoptosis, Bax, cleaved caspase-3, cleaved PARP, CTSB, AMPK and mTOR mRNA and protein levels	[64]
BMSCs	miR-29b-3p	Hypoxic-ischemic brain injury	in vivo: Cerebral ischemia induced by MCAO method in SD rats in vitro: OGD/R injury model in rat primary cortical neurons and BMECs	100 μg/kg/day intracerebroventricular stereotactic injection 2 h after the MCAO model, every day for 3 days	↑Bcl-2, VEGFA, VEGFR2, angiogenesis, and p-Akt/Akt; ↓Apoptotic cells, Bax, cleaved caspase-3, infarct volume, MVD, and PTEN;	[65]
Neuron, EC, NPC and ATC differentiated from H9 hES	_____	Ischemia	OGD-induced injury in H9 hES derived neurons	100 μg/mL	↑Neuronal survival rate, p-PI3K p85, p-Akt, p-mTOR, Bcl-2, and basal neuronal synaptic transmission; ↓Neuronal damage, p-AMPK, COX-2, iNOS, TNF-α, Bax, and cleaved caspase-3	[66]
EPCs	miR-126	Diabetic ischemic stroke	H/R and HG-induced injury in human astrocytes	3 × 10^9^ particles/mL	↓Cytotoxicity, ROS and lipid peroxidation	[67]

Abbreviations: ↑: increase or improvement; ↓: decrease or loss; AD: Alzheimer’s disease; ADSCs: adipose tissue-originated stromal cells; Akt: protein kinase B; AMPK: AMP-activated protein kinase; ATCs: astrocytes; Aβ: amyloid beta; BBB: blood–brain barrier; BDNF: brain-derived neurotrophic factor; BMECs: brain microvascular endothelial cells; BMSCs: bone marrow mesenchymal stem cells; CBF: cerebral blood flow; cMVD: cerebral vascular density; COX-2: cyclooxygenase-2; CREB: cAMP response element-binding protein; CRP: C-reactive protein; CST: corticospinal tract; CTSB: cathepsin B; CytC: cytochrome C; EBI: early brain injury; eNOS: endothelial nitric oxide synthesis; EPCs: endothelial progenitor cells; ERK1/2: extracellular signal-regulated kinase 1/2; ES: embryonic stem cell; GDF-15: growth differentiation factor-15; GSDMD-N: N-terminal of gasdermin D; H/R: hypoxia/reoxygenation; hES: human embryonic stem cell; HFD: high-fat diet; HG: high glucose; HucMSCs: human umbilical cord-derived MSCs; I/R: ischemia/reperfusion; ICH: intracerebral hemorrhage; ICMS: intracortical microstimulation; IFN-γ: interferon-γ; IGFBP5: insulin-like growth factor-binding protein 5; IL: interleukin; iNOS: inducible nitric oxide synthase; i.p.: intraperitoneal; IRAK1: interleukin-1 receptor-associated kinase 1; i.v.: intravenous; MAPK: mitogen-activated protein kinase; MCAO: middle cerebral artery occlusion; MCP-1: monocyte chemoattractant protein-1; miR: microRNA; mNSS: modified neurological severity score; MPO: myeloperoxidase; MSCs: multipotent mesenchymal stromal cells; mTOR: mammalian target of rapamycin; MVD: microvessel density; NDS: neurological deficit score; NFAT5: nuclear factor of activated T cells 5; NF-κB: nuclear factor-kappa B; NLRP3: nucleotide-binding domain and leucine-rich repeat family protein 3; nNOS: neuronal nitric oxide synthase; NPC: neural progenitor cell; NSC: neural stem cells; OGD/R: oxygen glucose deprivation/reperfusion; PARP: poly-adenosine diphosphate-ribose polymerase; p-CREB: phospho-cAMP response element-binding protein; p-GSK-3β: phosphorylated-glycogen synthase kinase-3 beta; PI3K: phosphoinositide-3-kinase; PTEN: phosphatase and tensin homolog; RASA1: Ras p21 protein activator 1; ROS: reactive oxygen species; SAH: subarachnoid hemorrhage; SC: stem cell; SD: Sprague–Dawley; SQSTM1: sequestosome 1; TLR7: toll-like receptor 7; TNF-α: tumor necrosis factor-α; TrkB: tropomyosin-related receptor kinase B; VD: vascular dementia; VEGFA: vascular endothelial growth factor A; VEGFR2: vascular endothelial growth factor receptor 2.

**Table 2 pharmaceutics-15-01006-t002:** Exosomes circumvent SCI and TBI via PI3K/Akt/mTOR and associated pathways.

Source of Exosomes	Cargo or Intermediate Molecule	Disease	Method (In Vitro/In Vivo)	Dose/Route of Administration	Mechanism of Actions and Outcomes	References
BMSCs	miR-26a	SCI	in vivo: SCI induced in SD rats in vitro: PC12 cells	200 μg in 200 μL PBS via tail vein injection 20 μg/mL for 48 h	↑Axonal regeneration, neurogenesis, functional recovery, BBB scores, MEP amplitudes, neurofilament density, Tuj-1, p-Akt, p-PI3K, p-mTOR, p-S6K, and p62; ↓Glial scarring, GFAP, astrocyte inflammation, autophagy, p-AMPK, p-ULK1, p-IKB, and p-p65; ↑Neurofilament generation, nerve regeneration, p-Akt, p-PI3K, and p-mTOR, ↓PTEN, autophagy, p-AMPK, p-ULK1, p-IKB and p-p65	[84]
BMSCs	miR-338-5p	SCI	in vivo: SCI induced in SD rats in vitro: H_2_O_2_-induced oxidative stress injury in PC12 cells	100 µg (50 µg microinjected to injured site +50 µg via the tail vein) in PBS at 5 min and 1 h after SCI, 100 μg of total protein	↑NF-M and GAP43; ↓MAG and GFAP; ↑Cell viability, SOD, NF-M, GAP43, Bcl-2, cAMP, Rap1, p-Akt, and p-PI3K; ↓Apoptosis, ROS, MAG, GFAP, Bax, cleaved caspase-3, and Cnr1	[85]
BMSCs under hypoxia	miR-216a-5p	SCI	in vivo: SCI induced in C57BL/6 mice in vitro: LPS-stimulated BV2 microglial cells and primary microglia	200 μg of total protein via tail vein injection 200 μg/mL	↑Functional recovery, BMS score, gait recovery, motor coordination, NeuN-positive neurons, MEP amplitudes, IL-4, IL-10, TGF-β, Arg1, CD206, YM1/2, and M1 to M2 polarization; ↓Neurofilament 200, lesion volume, iNOS, TNF-α, IL-1β, and IL-6; ↑IL-4, IL-10, Arg1, CD206, YM1/2, M1 to M2 polarization, TGF-β, p-Akt, and p-PI3K; ↓TNF-α, IL-1β, IL-6, iNOS, TLR4, p-P65, and MyD88;	[86]
GIT1-BMSCs	_____	SCI	in vivo: SCI induced in SD rats in vitro: Glutamate-induced injury model in neuronal cells	200 μg of total protein via tail vein injection 100 µg/mL	↑Nissl bodies, neural regeneration, BBB score, motor function, Bcl-2, and P-Akt; ↓Apoptosis, glial scar formation, TNF-α, IL-1β, IL-6, Bax, and cleaved caspase-3; ↑Bcl-2; ↓TUNEL-positive cells, neural apoptosis, Bax, cleaved caspase-3, and caspase-9	[87]
MSCs	_____	SCI	in vivo: SCI induced in SD rats in vitro: Astrocytes isolated from SCI rats	1 × 10^6^ in 200 μL PBS via tail vein injection, 5 × 10^4^	↑Functional recovery, MBP, BBB scores, Syn, and NeuN; ↓Lesion size, morphological phenomena, p65^+^ nuclei, A1 astrocytes, TNF-α, IL-1α, IL-1β, C3, GFAP, TUNEL-positive cells, p-IKBα, and p-p65	[88]
MSCs	miR-126	SCI	in vivo: SCI induced in SD rats in vitro: OGD injury model in HUVECs	100 µg of total protein in 0.5 mL PBS via tail vein injection 10 µg	↑Functional recovery, VEGF, angiogenesis, neurogenesis, blood vessels number and connectivity value, Bcl-2, NeuN, Sox2, and Nestin positive cells; ↓Lesion volume, incorrect steps, apoptosis, SPRED1, PIK3R2, Bax and cleaved caspase-3; ↑Angiogenesis and HUVECs migration; ↓SPRED1 and PIK3R2	[89]
Neuron-derived exosomes	miR-124-3p	SCI	in vivo: SCI induced in C57BL/6 mice in vitro: Primary microglial cultures and MCM, primary astrocyte cultures and primary neuronal cultures	200 μg of total protein in 200 μL of PBS via tail vein injection 200 μg/mL	↑Functional recovery, BMS score, gait recovery, motor coordination, MEP amplitudes, and hind limb alternation; ↓Forelimb dependence, axonal damage, TNF-α, IL-1α, IL-6, IL-1β, C1q, M1 microglia, iNOS, C3, and A1 astrocytes; ↑p-PI3K and p-Akt; ↓M1 microglia, iNOS, and p-P65	[90]
NSCs	FTY720	SCI	in vivo: SCI induced in SD rats in vitro: SCMECs hypoxic model	20 μg in 0.3 mL PBS via tail vein injection 20 μg/mL	↑Locomotor function, complete tissue structure, claudin-5, and Bcl-2; ↓Edema formation, inflammatory cell infiltration, SCI lesion, neuronal cell apoptosis, AQP4, and Bax; ↑ZO-1 and p-Akt ↓PTEN and SCMEC permeability	[91]
Primary SCDEs	_____	SCI	in vivo: SCI induced in mice in vitro: H_2_O_2_-induced injury in spinal cord astrocytes	0.1 μg/μL in 100 μL of DPBS via tail vein injection, three times a week for 4 weeks	↑TLR2, functional recovery, GFAP, 5-HT, BMS score, motor function, and neuron survival; ↓CSPGs deposition, p-PI3K/PI3K, and NF-κB; ↓p-PI3K/PI3K and NF-κB	[92]
Primary SCDEs	_____	SCI	in vivo: SCI induced in SD rats in vitro: H_2_O_2_-induced injury in PC12 cells	250 µL (0.1 µg/µL) in DPBS via tail vein injection, three times a week for 4 weeks	↑Autophagy, motor function, myelinated areas, NeuN, ChAT, LC3-1/2 and Beclin-1, P62; ↓Apoptosis, %cavity size and EGFR; ↑Neuronal survival, LC3-1/2, Beclin-1 and P62; ↓Apoptosis, EGFR, p-Akt, and p-mTOR	[93]
HUSC	ANGPTL3	SCI	in vivo: SCI induced in mice in vitro: HUVECs	200 μg in 200 μL of PBS via local intrathecal injection 200 μg	↑Spinal cord regeneration, locomotor function, BMS scores, sensory improvement, MEP amplitudes, angiogenesis, vessel volume fraction, vascular segment and bifurcation numbers; ↓Latent period and lesion cavities area; ↑Proliferation rate, cell migration, tube formation, angiogenic activities, p-Akt, and p-PI3K	[94]
hUC-MSCs	miR-199a-3p/145-5p	SCI	in vivo: SCI induced in SD rats in vitro: LPS-induced injury in PC12 cells	20 μg/mL	↑TrkA, locomotor function, and BBB score; ↓Lesion size, inflammation, and apoptotic cells ↑Cell viability, NF-H, β-tubulin-III, Neu-N, p-Akt, p-Erk, neurite outgrowth, and TrkA; ↓Cblb and Cbl expression;	[95]
PMs	_____	SCI	in vivo: SCI induced in SD rats in vitro: BV2 cells cultured in DMEM culture medium	20 and 200 μg/mL via tail vein injection 20 and 200 μg/mL	↑BBB score, inclined plate angle, Nissl-positive cells, IL-4, IL-10, and IL-13; ↓Tissue damage, IL-1β, IL-6, and TNF-α; ↑IL-10, CD206, CD163, Arg-1, LC3-II/Ⅰ, and Beclin-1; ↓p62, mTOR, and Akt protein level;	[96]
Primary microglia	Resveratrol	SCI	in vivo: SCI induced in SD rats in vitro: Mechanical injury model in primary spinal cord neurons	0.2 mL of PBS suspension of 40 µM for 14 days	↑Muscle tension, foot functional movements, BBB scores, motor function, neuron natural morphology and number, LC3B-positive cells, Beclin-1, and p-PI3K; ↓TUNEL-positive neurons, cleaved caspase-3, and apoptosis;	[97]
Pericytes	_____	SCI	in vivo: SCI induced in ICR mice in vitro: SCMECs hypoxic model	20 µg in 0.3 mL PBS via tail vein injection 20 µg/mL	↑Locomotor function, complete tissue structure, myelin sheath, Nissl body morphology and number, blood flow, Bcl-2, local microvascular disturbances, mean flux, and claudin-5; ↓Inflammatory cell infiltration, TUNEL-positive cells, Bax, HIF-1α, BSCB disruption, edema formation, MMP-2, and AQP4; ↑ZO-1 and p-Akt; ↓PTEN;	[98]
ECFCs	_____	TBI	in vivo: TBI induced in male C57BL/6 mice in vitro: RBMEC hypoxia injury model	4×10^6^ cell equivalents via the tail vein at 2 h after TBI,5 and 10 μg/mL	↑Neurological functional recovery, TJ, p-Akt, CLN5, occludin, and ZO-1; ↓Brain edema, PTEN, MMP-9, BBB permeability, EB dye extravasation, and TJ degradation	[101]
Microglia	miR-124-3p	TBI	in vivo: (r)TBI male C57BL/6 mouse model in vitro: Scratch injury model in pure cortical neurons	30 μg via tail vein, 3 × 10^8^ exosomes	↑Neurologic outcomes, neurite outgrowth and IL-10; ↓PDE4B, IL-1β, IL-6, TNF-α, mTOR signaling, neuronal inflammation, PDE4B, p-4E-BP1, and p-P70S6K;	[102]
Human ADSC	_____	TBI	in vivo: Weight-drop-induced TBI in male SD rats in vitro: LPS-induced inflammatory model	20 μg total protein per rat, 2.0 × 10^10^ particles/mL, intracerebroventricular injection	↑Functional recovery, neurogenesis, M1 to M2 microglial polarization, IGF1, arginase1, CD206 and IL-10; ↓Neuroinflammation, neuronal apoptosis, hippocampal neurogenesis, CD68+ activated microglia/macrophages, mNSS score, TNF-α, iNOS, IL-1α, IL-1β, IL-6, MCP-1, CCL2, CCL3, CCL5, morphological transformation, p-P38, p-IKKαβ, p-IKBα, p-P68, NF-κB, and P38/MAPK activation	[103]

Abbreviations: ↑: increase or improve; ↓: decrease or loss; Akt: protein kinase B; AMPK: AMP-activated protein kinase; AQP4: aquaporin-4; BAX: bcl-2-associated X protein; BBB: Basso, Beattie and Bresnahan; Bcl-2: B-cell lymphoma 2; BMS: Basso mouse scale; BMSCs: bone marrow mesenchymal stem cells; C1q: complement component 1q; C3: complement component 3; cAMP: cyclic adenosine monophosphate; CCL2: chemokine ligand 2; ChAT: choline acetyltransferase; Cnr1: cannabinoid receptor 1; CSPG: chondroitin sulfate proteoglycan; ECFCs: endothelial colony-forming cells; EGFR: epidermal growth factor receptor; GAP43: growth-associated protein-43; GFAP: glial fibrillary acidic protein; GIT1: G protein-coupled receptor kinase 2 interacting protein 1; H_2_O_2_: hydrogen peroxide; hUC-MSCs: human umbilical cord mesenchymal stem cells; HUSC: human urine stem cell; HUVECs: human umbilical venous endothelial cells; IGF1: insulin-like growth factor 1; IKBα: NF-kappa-B inhibitor alpha; IL: interleukin; iNOS: inducible nitric oxide synthase; i.p.: intraperitoneal; i.v.: intravenous; LPS: lipopolysaccharide; MAG: myelin associated glycoprotein; MAPK: mitogen-activated protein kinase; MBP: myelin basic protein; MCM: microglia-conditioned medium; MCP-1: monocyte chemoattractant protein-1MEP: motor-evoked potential; miR: microRNA; MMP-2: matrix metalloproteinase-2; MSCs: mesenchymal stem cells; NeuN: neuronal nuclei; NF-M: neuro filament-M; NSCs: neural stem cells; PDE4B: phosphodiesterase 4B; PI3K: phosphoinositide 3-kinase; PIK3R2: phosphoinositide-3-kinase regulatory subunit 2; PMs: peripheral macrophages; PTEN: phosphatase and tensin homolog; ROS: reactive oxygen species; SCDEs: Schwann cell-derived exosomes; SCI: spinal cord injury; SCMECs: spinal cord microvascular endothelial cells; SD: Sprague-Dawley; SOD: superoxide dismutase; SPRED1: sprouty-related EVH1 domain-containing protein 1; Syn: synaptophysin; TBI: traumatic brain injury; TGF-β: transforming growth factor-β; TJ: tight junction; TLR4: toll-like receptor 4; TNF-α: tumor necrosis factor-alpha; Tuj-1: β-tubulin-3; VEGF: vascular endothelial growth factor; ZO-1: zonula occludens-1.

**Table 3 pharmaceutics-15-01006-t003:** Exosomes circumvent other neurological diseases via PI3K/Akt/mTOR and associated pathways.

Source of Exosomes	Cargo or Intermediate Molecule	Disease	Method (In Vitro/In Vivo)	Dose/Route of Administration	Mechanism of Actions and Outcomes	References
ADSC	_____	ALS	in vitro: H_2_O_2_-induced injury in NSC-34 (G93A) cells	0.2 µg/mL, corresponding to 6–8 × 10^5^ particles/mL	↑Phospho-Akt, SOD1 gene, Bcl-2 α, and cell viability; ↓Cleaved caspase 3, Bax, and apoptosis;	[108]
Fibroblast	_____	CNS injury	in vitro: Cultured adult rat DRGs and RGCs	50 ng/mL	↑Cell survival, neurite growth, axonal growth, and pS6K; ↓CSPG and Wnt10b;	[109]
Fibroblast	miR-673-5p	Peripheral neuron myelinati on	in vivo: One-day-old newborn rats in vitro: Schwann cells	5 nmol/rat every 2 days via hypodermic injection	↑Myelin gene expression, mTORC1, SREBP2, Hmgcr, phosphatidylcholine, phosphatidylethanolamine, phosphatidylserine, diacyl glycerol, cholesterol, myelin sheath, myelinated axons, and myelin lamellae; ↓Tsc2 expression;	[110]
MSCs	miR-17-92	Axonal growth	in vitro: Primary cortical neurons under CSPG conditions	3 × 10^8^/300 µL and 3 × 10^9^/300 µL	↑Axonal growth, distal axons length, p-mTOR, and p-GSK-3β; ↓PTEN;	[111]
ADSCs	_____	Neural injury	in vitro: LPS-induced injury in SH-SY5Y and BV-2 cells	50 μg/mL	↑Cell viability; ↓Neuroinflammation, microglia cells, TNF-α, IL-1β, IL-6, COX-2, iNOS, cytotoxicity, p-P38, p-P65, p-ERK, and p-JNK;	[112]
hAMS	_____	Neural injury	in vitro: Neural injury induced by glutamate in PC12 cells	100 ng/mL	↑Cell survival and PI3K/Akt signaling activating;	[113]
MSCs	_____	ONC injury	in vivo: ONC-induced injury in SD rats	3 × 10^9^/5 μL, intravitreal injection	↑RGCs survival, IL-10, Bcl-2, and p-Akt; ↓IL-1β, IL-6, IL-8, MCP-1, Bax, TNF-α, cleaved caspase-3, and apoptosis;	[114]

Abbreviations: ↑: increase or improve; ↓: decrease or loss; ADSC: adipose-derived stem cells; Akt: protein kinase B; ALS: amyotrophic lateral sclerosis; CNS: central nervous system; COX-2: cyclooxygenase-2; CSPG: chondroitin sulfate proteoglycan; DRGs: dorsal root ganglia; ERK: extracellular signal-regulated kinase; GSK-3β: glycogen synthase kinase-3 beta; hAMS: human adipose-derived mesenchymal stem cells; Hmgcr: 3-hydroxy-3methylglutaryl coenzyme A; IL: interleukin; iNOS: inducible nitric oxide synthase; JNK: c-Jun N-terminal kinase; LPS: lipopolysaccharide; MCP-1: monocyte chemoattractant protein-1; MSCs: mesenchymal stem cells; mTOR: mammalian target of rapamycin; mTORC1: mechanistic target of the rapamycin complex 1; ONC: optic nerve crush; PTEN: phosphatase and tensin homolog; RGCs: retinal ganglion cells; SREBP2: sterol-regulatory element binding protein 2; TNF-α: tumor necrosis factor-α.

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
