# Peer review of "The Exosome-Mediated PI3K/Akt/mTOR Signaling Pathway in Neurological Diseases"

_pharmaceutics, 2023, doi:10.3390/pharmaceutics15031006_

Round 1
Reviewer 1 Report
Amin Iranpanah and colleagues systemically summarized the role of exosome mediated P13K/Akt/mTOR signaling pathway in neurodegenerative diseases (NDDs), including Alzheimer’s disease, stroke, spinal cord injury, traumatic brain injury, and other type of NDDs. They discussed the therapeutic effect of exosomes in regulating inflammation, oxidative stress and apoptosis in NDDs through P13K/Akt/mTOR signaling pathway. The topic is well chosen and the review is well organized and comprehensive. I have the following comments and suggestions:
1. Please add the dose of exosomes used in the studies that shows in Table 1, Table 2 and Table 3.
2. Please add the route of administration of the exosomes for the in vivo studies shows in Table 1, Table 2, and Table 3.
3. Please briefly discuss the challenges of exosomes application in the “conclusions and future perspective” section.
Author Response
Reviewer # 1
Overall comment: Amin Iranpanah and colleagues systemically summarized the role of exosome mediated P13K/Akt/mTOR signaling pathway in neurodegenerative diseases (NDDs), including Alzheimer’s disease, stroke, spinal cord injury, traumatic brain injury, and other type of NDDs. They discussed the therapeutic effect of exosomes in regulating inflammation, oxidative stress and apoptosis in NDDs through P13K/Akt/mTOR signaling pathway. The topic is well chosen and the review is well organized and comprehensive. I have the following comments and suggestions:
Response: We appreciate respected reviewer for positive consideration of our manuscript towards publication. The “point-by-point” and itemized responses to each comment are applied.
Comment 1: Please add the dose of exosomes used in the studies that shows in Table 1, Table 2 and Table 3.
Response: Thank you for the comment. The dose of administration of exosomes added to reports presented in Tables 1, 2 and 3 (Table Column: Dose/Route of administration).
Comment 2: Please add the route of administration of the exosomes for the in vivo studies shows in Table 1, Table 2, and Table 3.
Response: It was a critical comment. The route of administration of exosomes added to reports presented in tables 1, 2 and 3 (Table Column: Dose/Route of administration).
Comment 3: Please briefly discuss the challenges of exosomes application in the “conclusions and future perspective” section.
Response: We appreciate the comment. Challenges of exosomes briefly added in the section of “conclusions and future perspective”.
Regards
Reviewer 2 Report
This reviewer appreciates the attempt the authors have made to do a thorough literature review of the role exosomes play in the pathogenesis, progression and potential alleviation of symptoms of various neurodegenerative disorders. However, the scope of this is too broad and the review comes off as a very verbose summarization of the conclusions of each of the papers they shortlisted but without offering the authors’ own critical insight into the results of the articles chosen. A good literature review should be a lot more than just a summary of the results of each paper. There are also language issues at multiple instances throughout the manuscript which makes it difficult to follow what the authors are attempting to say and in other instances, the meaning of the sentence is contrary to what the cited study suggests. I would make the following suggestions to improve this manuscript:
· Narrow the scope of the review and focus on either exosomes derived from cells of the CNS OR one or two NDDs instead of trying to fit ALL this information into one review.
· With a narrowed scope, the authors should delve into more detail of the fewer papers they will now have to discuss and can therefore offer their own perspectives on the studies they are reviewing, such as, what do those studies achieve, how did their results address the gaps in knowledge regarding the selected NDDs, what did the study lack, what questions remained unanswered.
· It would also be beneficial to include a more in depth discussion section focusing on the therapeutic potential of exosomes for the selected NDDs as well as experimental suggestions for how that may be accomplished.
The goal of a good review should be to educate the reader in depth on a smaller focused topic. I hope the authors will take these suggestions and improve their manuscript.
Author Response
Reviewer # 2
Overall comment: This reviewer appreciates the attempt the authors have made to do a thorough literature review of the role exosomes play in the pathogenesis, progression and potential alleviation of symptoms of various neurodegenerative disorders. However, the scope of this is too broad and the review comes off as a very verbose summarization of the conclusions of each of the papers they shortlisted but without offering the authors’ own critical insight into the results of the articles chosen. A good literature review should be a lot more than just a summary of the results of each paper. There are also language issues at multiple instances throughout the manuscript which makes it difficult to follow what the authors are attempting to say and in other instances, the meaning of the sentence is contrary to what the cited study suggests. I would make the following suggestions to improve this manuscript:
Response: We appreciate respected reviewer for the valuable comments which are responded “point-by-point” and itemized. The language editing done by an expert.
Comment 1: Narrow the scope of the review and focus on either exosome derived from cells of the CNS OR one or two NDDs instead of trying to fit ALL this information into one review.
Response: We appreciate the valuable comment of respected reviewer. However, since there was no previous review to reveal the pivotal role of PI3K/Akt/mTOR pathway and exosomes modulatory effects on the overall NDDs/neurological disorders, in the present review we highlighted those less-traveled ways. In the light of your valuable comment, we will specially focus on major NDDs in future reports.
Comment 2: With a narrowed scope, the authors should delve into more detail of the fewer papers they will now have to discuss and can therefore offer their own perspectives on the studies they are reviewing, such as, what do those studies achieve, how did their results address the gaps in knowledge regarding the selected NDDs, what did the study lack, what questions remained unanswered.
Response: According to your helpful and valuable comments, our views on the studies being reviewed and challenges of exosomes briefly added to the conclusion section of manuscript. In addition, the dose of exosomes used in the studies and the route of administration of the exosomes added to the manuscript. At the end of each section, a brief description done on the overall results of included reports. Additionally, it was a critical goal to reveal the pivotal role of PI3K/Akt/mTOR pathway and exosomes modulatory effects on the overall NDDs/neurological disorders, in the present review, since there was no previous review on the topic (as presented in the response of comment 1).
Comment 3: It would also be beneficial to include a more in-depth discussion section focusing on the therapeutic potential of exosomes for the selected NDDs as well as experimental suggestions for how that may be accomplished.
Response: Thank you for the comment. The more in-depth discussion was provided.
Regards
Reviewer 3 Report
In this review, the authors reported literature regarding the essential functions of exosomes as pivotal therapeutic targets in the NDDs pathogenesis passing through PI3K/Akt/mTOR pathway.
The topic is interesting, however, there are critical major points to address:
1- First of all, it is not clear whether the authors performed a systematic review or not. If yes, the authors should carefully follow the guidelines described in Page et al 2021 (The PRISMA 2020 statement: an updated guideline for reporting systematic reviews). To complete and give an added value to their work, the authors should perform also a quality assessment of the works selected for the systematic review.
2- A further key point of this work is that the authors should choose clearly which kind of pathologies they wanted to include in the neurodegenerative disorders. Did they select neurological diseases? If yes, specify. In the MeSH selection they included “depression”, “epilepsy”, “stroke” and “autism” that are not neurodegenerative disorders. Moreover, in the table 3 the authors inserted under “diseases” axonal growth, neural injury, ONC and CNS injury and Peripheral neuron myelination that they are not neurodegenerative diseases. It is also important to specify what the authors meant with “miscellaneous NDD”.
3- The paragraph “2. Axial role of PI3k/Akt/mTOR signaling pathway in neuroprotective responses” should be integrated in the introduction to be linked with the main aim of the review. If the introduction becomes too long, a division in different subparagraphs could be recommended.
4- Consequently, the aim of the paper at the end of the introduction must be clear and the authors should emphasize the added value and the originality of this review.
Minor points
- check for acronyms that are sometimes not described, while other times they are described but not used.
Author Response
Reviewer # 3
Overall comment: In this review, the authors reported literature regarding the essential functions of exosomes as pivotal therapeutic targets in the NDDs pathogenesis passing through PI3K/Akt/mTOR pathway. The topic is interesting, however, there are critical major points to address:
Response: We appreciate respected reviewer for positive consideration of our manuscript towards publication. The “point-by-point” and itemized responses to each comment are applied.
Comment 1: First of all, it is not clear whether the authors performed a systematic review or not. If yes, the authors should carefully follow the guidelines described in Page et al 2021 (The PRISMA 2020 statement: an updated guideline for reporting systematic reviews). To complete and give an added value to their work, the authors should perform also a quality assessment of the works selected for the systematic review.
Response: Thank you for the comment. The current systematic review study was done on the basis of PRISMA criteria. The flow diagram of the process of literature search and selection of relevant reports was shown in figure 1. However, this is not a Meta-Analysis to do a quality assessment of the included articles.
Comment 2: A further key point of this work is that the authors should choose clearly which kind of pathologies they wanted to include in the neurodegenerative disorders. Did they select neurological diseases? If yes, specify. In the MeSH selection they included “depression”, “epilepsy”, “stroke” and “autism” that are not neurodegenerative disorders. Moreover, in the table 3 the authors inserted under “diseases” axonal growth, neural injury, ONC and CNS injury and Peripheral neuron myelination that they are not neurodegenerative diseases. It is also important to specify what the authors meant with “miscellaneous NDD”.
Response: Neurodegenerative diseases and neurological disorders are included in the manuscript and associated corrections done. “Miscellaneous NDDs” is also corrected as “other neurological disorders”.
Comment 3: The paragraph “2. Axial role of PI3k/Akt/mTOR signaling pathway in neuroprotective responses” should be integrated in the introduction to be linked with the main aim of the review. If the introduction becomes too long, a division in different subparagraphs could be recommended.
Response: It is a critical comment. In the light of your comment, the section “2. Axial role of PI3k/Akt/mTOR signaling pathway in neuroprotective responses”, integrated in the introduction.
Comment 4: Consequently, the aim of the paper at the end of the introduction must be clear and the authors should emphasize the added value and the originality of this review.
Response: We appreciate the comment. The aim of study and related added value completed at the end of introduction. It is also highlighted that “no review revealed the effects of exosome-mediated PI3K/Akt/mTOR signaling pathway in NDDs”
Comment 5: Minor points -check for acronyms that are sometimes not described, while other times they are described but not used.
Response: Thank you for the comment. Revision is done in light of your comment.
Regards
Round 2
Reviewer 2 Report
While language editing has been done and the flow of language is better, there are many instances wherein the language in certain sentences inaccurately represents the facts. Further language editing is required, perhaps by a scientific language editor with experience in neuroscience manuscripts, in order to make sure the correct message is conveyed.
I do not see what in depth discussion was added or where the authors included suggestions previously made about the different studies they highlighted. While it is admirable to want to be the first to present a review on an exciting new field, in order for it to be useful to the reader, it must include detailed information on the subject and not just summarize the main findings of articles. While the authors have added their views on the general shortcoming of exosomes, there is no new information added specifically on the studies they chose to highlight for each NDD. I would again suggest to pick some of the NDDs and delve into further detail about the role of exosomes in those particular NDDs and go into more details about the studies, their findings and their shortcomings. Present your own thoughts on what experiments could be done to improve the shortcomings for using exosomes as therapeutic agents. Generalized statements don't add any new/original insight for the reader. Nearly ever study needs further in vitro/ in vivo testing. Be specific about what you think could be done experimentally based on the knowledge you have gleaned from the various articles you read for this review.
Regarding the discussion and new additions, it seems like similar information has been added in different places in the manuscript. Lines 188-189; 302-306 and 479-485 are basically reiterations of some of the same points and even the language hasn't been altered much.
Author Response
Reviewer # 2
Comment 1: While language editing has been done and the flow of language is better, there are many instances wherein the language in certain sentences inaccurately represents the facts. Further language editing is required, perhaps by a scientific language editor with experience in neuroscience manuscripts, in order to make sure the correct message is conveyed.
Response: We did a major English editing by a scientific language editor with experience in neuroscience (highlighted in red).
Comment 2: I do not see what in depth discussion was added or where the authors included suggestions previously made about the different studies they highlighted. While it is admirable to want to be the first to present a review on an exciting new field, in order for it to be useful to the reader, it must include detailed information on the subject and not just summarize the main findings of articles. While the authors have added their views on the general shortcoming of exosomes, there is no new information added specifically on the studies they chose to highlight for each NDD. I would again suggest to pick some of the NDDs and delve into further detail about the role of exosomes in those particular NDDs and go into more details about the studies, their findings and their shortcomings. Present your own thoughts on what experiments could be done to improve the shortcomings for using exosomes as therapeutic agents. Generalized statements don't add any new/original insight for the reader. Nearly ever study needs further in vitro/ in vivo testing. Be specific about what you think could be done experimentally based on the knowledge you have gleaned from the various articles you read for this review. Regarding the discussion and new additions, it seems like similar information has been added in different places in the manuscript. Lines 188-189; 302-306 and 479-485 are basically reiterations of some of the same points and even the language hasn't been altered much.
Response: We added in-depth discussion at the end of main sections 3.1, 3.2, 3.3, 3.4. More details of the included studies are also presented in the text and tables. The discussion and conclusion sections are also completed in light of your comment (highlighted in red).
Regards
Reviewer 3 Report
The authors have adequately replied to all my comments and therefore the work is now eligible for publication.
Author Response
Thanks for your acceptance our work
Round 3
Reviewer 2 Report
The review reads much better now with the scientific editing and incorporating more details at the end of each section as well as in the conclusion.